HRV-derived data similarity and distribution index based on ensemble neural network for measuring depth of anaesthesia

Liu Quan 1 2
Ma Li 1 2
Chiu Ren-Chun 3
Fan Shou-Zen 4
Abbod Maysam F. 5
Shieh Jiann-Shing jsshieh@saturn.yzu.edu.tw 3
1 Key Laboratory of Fiber Optic Sensing Technology and Information Processing (Wuhan University of Technology), Ministry of Education , Wuhan , China
2 School of Information Engineering, Wuhan University of Technology , Wuhan , China
3 Department of Mechanical Engineering and Innovation Center for Big Data and Digital Convergence, Yuan Ze University , Taoyuan , Taiwan
4 Department of Anesthesiology, National Taiwan University , Taipei , Taiwan
5 Department of Electronic and Computer Engineering, Brunel University London , Uxbridge , United Kingdom
Boonstra Tjeerd
Electronic publication date: 2017 Nov 16
Publication date: 2017
Volume: 5
Electronic Location ID: e4067
Received 2017 Jul 31; Accepted 2017 Oct 29
Copyright: ©2017 Liu et al.
Copyright year: 2017
Copyright holder: Liu et al.
License: This is an open access article distributed under the terms of the Creative Commons Attribution License, which permits unrestricted use, distribution, reproduction and adaptation in any medium and for any purpose provided that it is properly attributed. For attribution, the original author(s), title, publication source (PeerJ) and either DOI or URL of the article must be cited.
License URL: https://creativecommons.org/licenses/by/4.0/

Keywords: Heart rate variability, Depth of anesthesia, Similarity and distribution index, Artificial neural network, Expert assessment of consciousness level

Funding: National Science Council NSC102-2911-I-008-001 Innovation Center for Big Data and Digital Convergence, Yuan Ze University, Taiwan Wuhan University of Technology international exchange program 2015-JL-012 National Natural Science Foundation of China 51475342 51675389 This research is supported by the Center for Dynamical Biomarkers and Translational Medicine, National Central University, Taiwan, which is sponsored by National Science Council (Grant Number: NSC102-2911-I-008-001). This research is also supported by Innovation Center for Big Data and Digital Convergence, Yuan Ze University, Taiwan. Additional funding comes from Wuhan University of Technology international exchange program (Grant Number: 2015-JL-012) and National Natural Science Foundation of China (Grant Number: 51475342, 51675389). The funders had no role in study design, data collection and analysis, decision to publish, or preparation of the manuscript.

==============================
Evaluation of depth of anaesthesia (DoA) is critical in clinical surgery. Indices derived from electroencephalogram (EEG) are currently widely used to quantify DoA. However, there are known to be inaccurate under certain conditions; therefore, experienced anaesthesiologists rely on the monitoring of vital signs such as body temperature, pulse rate, respiration rate, and blood pressure to control the procedure. Because of the lack of an ideal approach for quantifying level of consciousness, studies have been conducted to develop improved methods of measuring DoA. In this study, a short-term index known as the similarity and distribution index (SDI) is proposed. The SDI is generated using heart rate variability (HRV) in the time domain and is based on observations of data distribution differences between two consecutive 32 s HRV data segments. A comparison between SDI results and expert assessments of consciousness level revealed that the SDI has strong correlation with anaesthetic depth. To optimise the effect, artificial neural network (ANN) models were constructed to fit the SDI, and ANN blind cross-validation was conducted to overcome random errors and overfitting problems. An ensemble ANN was then employed and was discovered to provide favourable DoA assessment in comparison with commonly used Bispectral Index. This study demonstrated the effectiveness of this method of DoA assessment, and the results imply that it is feasible and meaningful to use the SDI to measure DoA with the additional use of other measurement methods, if appropriate.

Introduction

Anaesthesia is a significantly important procedure used in almost all surgery (Lan et al., 2012; Schwartz et al., 2010). General anaesthesia is a drug-induced and reversible condition that has specific behavioural and physiological effects such as unconsciousness, analgesia, and akinesia. Clinically and practically, routine observations such as those of heart rate, respiration, blood pressure, lacrimation, and sweating are used to assist doctors in smoothly controlling and safely managing anaesthesia. Nevertheless, patients recovering from general anaesthesia can experience significant clinical challenges, including airway and oxygenation problems, emergence delirium (Lepouse et al., 2006), cognitive dysfunction (Saczynski et al., 2012), and delayed emergence, and the elderly are particularly at risk of stroke and heart attack (Neumar et al., 2008). Accurate monitoring of the depth of anaesthesia (DoA) would thus contribute to improvements in the safety and quality of anaesthesia use and would provide a superior experience for patients.

A state of general anaesthesia is produced by anaesthetics that act on the spinal cord and the stem and cortex of the brain (Brown, Purdon & Van Dort, 2011; Ching & Brown, 2014); monitoring of electroencephalogram (EEG) patterns is therefore useful (Niedermeyer & Da Silva, 2005). The two main indices derived from an EEG pattern are the bispectral index (BIS) (Aspect Medical Systems, Newton, MA, USA) (Rosow & Manberg, 2001) and entropy (GE Healthcare, Helsinki, Finland) (Viertiö-Oja et al., 2004); the former is obtained by calculating adjustable weights on the power spectrum, the burst suppression pattern, and the bispectrum of EEG data, whereas the latter is constructed by associating the data degree of disorder (entropy) with the consciousness state of patients (Liang et al., 2015; Viertiö-Oja et al., 2004). Although EEG-based spectral indices have been applied commercially for nearly 20 years, they are still not part of standard anaesthesiology practice (Purdon et al., 2015), and the reasons for this are complex. First, these indices were developed from adult patient cohorts, and are not strictly relevant to infants or younger patients, thereby providing lower accuracy (Samarkandi, 2006), and second, the indices cannot generate precise DoA measurements for certain drugs, especially when ketamine and nitrous dioxide are used (Avidan et al., 2008; Sleigh & Barnard, 2004). In addition, EEG signals are sensitive to noise, and therefore more complex algorithms and resources for noise filtering are required. Moreover, using disposable EEG electrodes is much more expensive than using other physiological signal sensors.

To overcome some of these disadvantages and provide alternatives to EEG-based solutions (Ahmed et al., 2011), it is crucial to pursue new ideas to support mainstream methods. In this respect, the electrocardiogram (ECG) provides important clinical physiological signals and is highly recommended for continuous monitoring and ensuring international standards for the safe practice of anaesthesia (Merry et al., 2010). Different anaesthetics affect the QT interval of an ECG during anaesthetic induction (Oji et al., 2012), and rhythmic-to-non-rhythmic observations from the ECG can provide anaesthetic information (Lin , 2015). In addition, heart rate variability (HRV), related to autonomic regulation, is strongly affected by general anaesthesia (Hsu et al., 2012) and varies with respect to differing anaesthetic procedures used (Billman, 2013; Mazzeo et al., 2011); therefore, heartbeat dynamics are highly correlated with a loss of consciousness (Citi et al., 2012). Furthermore, ECG signals are more stable than EEG signals, which means that ECG is more resistant to noise even when cheap electrode sensors are used. HRV analysis thus can be used to estimate DoA. Moreover, interindividual variation is normal and is influenced by age, weight, and life habits, which means that the ECG-derived index more specifically reflects an individual’s anaesthetic state than EEG-based indices that assume one index value indicates the same consciousness level for all anaesthetics and patients (Purdon et al., 2015). Performing DoA research based on the HRV is thus valuable. However, it is important to guarantee that the ECG is free of artefacts and the ECG waveform (Q R S T waveform) is accurately recognised; otherwise, incorrect variation properties may ultimately be obtained, resulting in an incorrect R–R interval distribution.

An artificial neural network (ANN) is an advanced modelling tool used in statistics, machine learning, and cognitive science (Alpaydin, 2014; Kriegeskorte, 2015). This bio-inspired method supports self-learning from complex data by organizing training pattern set and resultant errors between the preferred output and the subsequent network output. It has the great ability of non-linear, distributed, local, and parallel processing and adaptation and one of the most often used models in engineering applications. An ensemble artificial neural network (EANN) comprises multiple models and combines them to produce the desired output, as opposed to using a single model (Kourentzes, Barrow & Crone, 2014; Ripley & Ripley, 2001; Tay et al., 2013). Normally, an ensemble of models performs better than any individual model because average effects are obtained in ensemble models (Baraldi et al., 2013; Zhou, Wu & Tang, 2002). In summary, the neural network is a powerful and effective method for use in data regression and model optimisation of nonstationary data. In biomedical fields, neural networks play a crucial role in the analysis of complex physiological data (Amato et al., 2013).

This study aimed to optimise an indicator index, known as the similarity and distribution index (SDI), that is derived from measurements of HRV (Huang et al., 2008). The SDI is proposed to evaluate the DoA from ECG signals occurring in the time domain during routine surgery, and thus differs from the methods previously described herein, which are based on extracting EEG spectrum features in the frequency domain. The time domain parameter is calculated by measuring the similarity between the statistical distributions of R–R interval measurements in consecutive data segments. In this study, results obtained using the proposed method are compared with the expert assessment of consciousness level (EACL), which is determined using the average evaluation of five expert anaesthetists after data and patient observation. The model is then optimised by applying an EANN for estimating the DoA. Through SDI extraction in the time domain and EANN modelling targeting the EACL, results show that it is possible to predict the DoA throughout an entire surgery.

The remainder of this paper is divided into four sections. ‘Materials and Methods’ describes the general anaesthesia used, patient participants and data analysis methods employed; ‘Results’ presents the results of processing and comparisons with the EACL; ‘Discussion’ presents the discussion and study limitations; and the conclusion is provided in ‘Conclusions’.

Materials and Methods

Ethics statement

All studies were approved by the Research Ethics Committee, National Taiwan University Hospital (NTUH), Taiwan, and written informed consent was obtained from patients (No: 201302078RINC). During the experimental trial, the hospital endeavoured to ensure that all scheduled surgery was performed very well on time.

Standard anaesthetic procedure

Anaesthesia is essential during surgery, and its associated procedures are outlined in Fig. 1 (Cornelissen et al., 2015). Anaesthesia generally involves end-tidal gas concentration over time, and routine anaesthetic practice consists of four stages: consciousness, induction, maintenance, and emergence (recovery) (Merry et al., 2010). Prior to surgery, patients were required to take nil by mouth for at least 8 h. After the electrodes were placed, each patient received the volume of anaesthetic agents appropriate for the routine operation. Unconsciousness is usually induced by intravenous propofol, another analgesic drug (such as fentanyl), and a muscle relaxant medicine (such as nimbex). Gas anaesthetics (desflurane, sevoflurane) together with air and oxygen were used to maintain sedation for most patients after the mask had been placed, whereas propofol was employed in some cases. As the end of surgery approached, additional drugs were administrated (such as morphine and atropine). Table 1 summarises detailed information. General anaesthesia was performed safely during all stages by monitoring physiological signals, such as EEG, ECG, photoplethysmography (PPG), and intermittent vital signs of blood pressure (BP), heart rate (HR), pulse rate (PR), and pulse oximeter oxygen saturation (SpO2). If any of these observation signals underwent irregular changes, the anaesthetist adjusted the intraoperative standard anaesthesia machine correspondingly.

Figure 1 Anesthetic procedure.

Table 1 Patients clinical characteristics and demographics.

Values are means (SD). Some eligible subjects are excluded by reasons described in Fig. 2.

Parameters		
Age (year)	49.0 (12.5)	
Male gender (%)	16.4%, n = 18	
Height (cm)	158.7 (7.6)	
Weight (kg)	59.4 (12.7)	
BMI (kg m−2)	23.6 (4.9)	
Median duration of surgery (min)	120 (CI:113.9∼138.9)	
Anesthetic management		
Propofol induction (mg)	115.6 (34.3), n = 100	
Fentanyl induction (mg)	95.5 (41.4), n = 100	
Lidocaine induction (mg)	48.1 (6.5), n = 60	
Glycopymolfe induction (mg)	0.2 (0.04), n = 64	
Nimbex induction (mg)	9.5 (1.7), n = 50	
Xylocaine induction (mg)	44.5 (9.0), n = 33	
Rubine induction (mg)	0.2 (0.06), n = 32	
Maintenance drugs infusion rate	–	
Sevoflurane maintenance (%)	53.4%, n = 59	
Desflurane maintenance (%)	35.5%, n = 39	
Propofol maintenance (%)	29.1%, n = 32	
Additional drugs administrated when approaching the end of surgery	
Morphine (mg)	4.5 (2.3), n = 47	
Ketamine (mg)	29.8 (7.3), n = 25	
Atropine (mg)	1.1 (0.4), n = 49	
Vagostin (mg)	2.4 (0.2), n = 48	
Notes.

BMI body mass index

Data recording

ECG data acquired in this study were obtained from patients undergoing surgery at the NTUH using chest-mounted sensors and a MP60 anaesthetic monitor machine (Intellivue; Philips, Foster City, CA, USA). The machine was connected to a recording computer installed with real-time software developed by our research team using a Borland C+ + Builder 6 developing environment kit (Borland Company, C+ + version 6); this software collected data at a sampling rate of 500 Hz. The sampling rates of the EEG and PPG continuous waveforms were 128 Hz. Intermittent vital signs (such as BIS, HR, PR, BP and SPO2) were recorded every 5 s.

Figure 2 Study protocol.

In fact, patients before this collection period were consulted for their eligibility, dozens of cases were excluded for analysis such as technical and clinical reasons. The 110 remaining subjects are intact for four stages of analysis to evaluate depth of anaesthesia (DoA). Their demography information is shown in Table 1.

Clinical data collection

Prior to collecting data in this study, patients provided written consent for participation. Demographic and clinical data, including height, weight, age, gender, operation time, surgical procedure, and anaesthetic management, were acquired by hospital staff from anaesthesia recording sheets. Other data relating to the research procedure, such as body movement and electrotome operation, were recorded by the research team. Regular hospital recordings and specific research notes were then integrated to serve as auxiliary clinical information.

Patient participants

Patients scheduled for elective surgical procedures were recruited from the preoperative clinic at NTUH in 2015. Eligibility criteria related to age, consent, and specific operation type. Those ineligible for inclusion were either (1) under 22 years old, (2) diagnosed with a neurological or cardiovascular disorder, or (3) undergoing surgery involving local anaesthetic rather than general. The selection procedure is illustrated in Fig. 2. According to these criteria, hundreds of patients were eligible for inclusion. However, it was unfortunately not possible to obtain data for all eligible patients (technique failure, procedure interruption), and ultimately data for 110 patients were acquired. General parameter information was obtained for all 110 patients. However, anaesthetic drug management differed with respect to individuals, although propofol and fentanyl inductions were implemented for most patients (n = 100). The detailed characteristics of the participants are provided in Table 1.

ECG data preprocessing

Data conditioning

Data conditioning, or preprocessing, is critical for signal analysis for determining DoA and can overcome problems with compatibility and a lack of analysis in advance. It generally consists of data format conversion, noise removal, and data rearrangement. Due to limitations with data collection storage, an ASCII file format was used in this study. Prior to implementing the algorithm, data were transferred into a MATLAB workspace and the notch filter was then used to remove 60 Hz power line noise. All participant data sets were then manually inspected to determine specific segments of artefacts resulting in extremely abnormal QRS waveforms or ECG series saturation (for example, electrical artefacts caused by medical equipment or body movement), particularly for the R peak, which was previously impossible to recognise. Our algorithm was then applied to pre-processed data for further analysis.

EACL

It is common knowledge that no accurate standard index exists that is capable of symbolising a patient’s anaesthetic state during clinical surgery. Therefore, five experienced anaesthesiologists were asked to plot subjective scores relating to ‘state of anaesthetic depth’ versus time, based on the data recordings referred to in the previous section and their own rich clinical experience. These scores thus represented an EACL. Criteria determined by the five anaesthesiologists with respect to their assessments of consciousness level were based on both their clinical practice knowledge and supporting information recorded by two research nurses. Any clinical events and signs potentially related to DoA were carefully recorded. Recorded information included (i) intermittent vital signs (such as HR, BP, SPO2); (ii) anaesthetic events, including induction of anaesthesia, tracheal intubation and extubation, the addition of muscle relaxant reversal drugs, and managing airway suction; (iii) surgical events, including the start and end of surgical procedure and the occurrence of any specific noxious stimulus; (iv) clinical signs of the patient, including any types of movement and unusual responses and arousability during induction and emergence from anaesthesia; and (v) any other events that were considered relevant, such as patient demography.

Figure 3 Flowchart design of expert assessment of consciousness level (EACL).

Recordings are clinically related BP, HR, SPO2 and drug administration records; assessments are done by five experienced experts by plotting the DoA curves with range from 0 to 100. After using ANSYS to digitalize the curve value, we obtained the final gold standard by averaging the five doctors’ assessments. EACL: expert of assessment of consciousness level.

Figure 4 One representative of EACL.

From (A) to (E), it is the five doctors’ assessments, respectively; the final one (F) is the gold standard: EACL. The Red solid line is the mean value, the two green dashed line is mean ± std.

Figure 5 Similarity and distribution index (SDI) definition protocol.

ECG denotes step 1, R (n) means step 2. Step 3 includes D (n) and histogram. The histogram distribution is used for SDI computation.

Based on these criteria (Liu et al., 2015), the assessment procedure used in this study to score DoA (Fig. 3) is described as follows. First, research nurses continually observed each patient’s state to record the information described above. Each anaesthesiologist then made a continuous assessment and noted changes in ‘the state of anaesthetic depth’ of patients during the entire operation, based on hospital formal anaesthesia records. To maintain consistency with the BIS, scoring used the range of 0–100, from brain dead to fully awake (a score of 40–65 represents an appropriate level of anaesthesia during surgery). Finally, because original assessments were drawn by hand, the results were digitised using web-based software (webplotdigitizer; ANSYS, Canonsburg, PA, USA) (Dorogovtsev & Mendes, 2013) and resampled every 5 s using MATLAB interpolation to ensure concurrence with the BIS index. The result was then considered to be an EACL. However, because the experience of each anaesthesiologist differed with respect to subjective EACL standards, and to minimise the consciousness level error as much as possible, the data obtained from the five anaesthesiologists were averaged. Figure 4 shows one EACL case example from the five doctors and the mean value of the five scores, where it is evident that the mean value better represents absolute DoA.

Data analysis

SDI definition of HRV

SDI protocol.

The SDI is based on HRV recorded in ECG data. The SDI is a time domain parameter index representing the degree of similarity between consecutive data segments and is obtained by computing the statistical distribution of the R–R interval variability difference. Figure 5 shows details of the entire procedure used to compute the SDI from ECG recordings. The steps involved are as follows:

Step 1. Extract the R peak of the ECG signal to obtain the instantaneous R–R interval, Rn. Resample the data using the commonly used algorithm of Berger to 4 Hz (Berger et al., 1986).

Step 2. Calculate the difference between two consecutive heartbeat intervals: (1) Dn=Rn+1−Rnn=1,2,3…

Step 3. Choose any time point, t, and then select a data block, where the data block contains M data points. Compare the statistical distribution of consecutive blocks, one from D(t − M + 1) to D(t), the other from D(t + 1) to D(t + M). Distribution histograms of both data blocks are generated using the same cell size. The relative frequency of the Dn value of the ith cell of the histogram is denoted P1(i) for the first data block and P2(i) for the second. Determination of the cell number is described in ‘Data analysis’ part B below. For example, in the first data block, the data value range is 0 to 0.5 s if 100 cells are chosen, and the cell width should be set as 0.005 s. This means that P1(1) denotes the relative frequency between 0 and 0.005; that is, P11= relative frequency 0<Dn<0.005,P12= relative frequency 0.005<Dn<0.010 and so on. This is the same for the second data block.

Step 4. After multiplying the relative frequency of corresponding cells in the histograms of both data blocks, the sum of the product value in all cells is the SDI, as calculated using the following equation: (2) SDI=1−∑i=1nP1i ∗P2i×100,

where n is the number of cells and P1(i), P2(i) are the relative frequencies of each cell in the histograms of data blocks 1 and 2, respectively. Theoretically, high similarity between the distribution features of ECG data means that patients are in a stable physical condition during surgery and that they are under a state of anaesthesia with high values of P1 × P2. When the sum is deducted by 1 and shows a lower SDI, the DoA is deeper. When the sum is multiplied by 100, the index value ranges from 0 to 100 and is consistent with clinically recognised consciousness levels, such as BIS values that range from 0 (deep coma) to 100 (awake state), thus making it easier to determine the DoA.

Implication of SDI value.

Mathematically, the SDI is obtained from measuring features of the statistical distribution between two consecutive data segments. For a stable HR pattern, the consecutive data segments should have high similarity and a histogram will show a consistent distribution when P1i and P2i fluctuate simultaneously. Under the condition of Eq. (2), the SDI is lower in this situation; therefore, a higher SDI symbolises a much more variable HR, which occurs frequently when a patient is awake or under minimal anaesthesia. In this instance, the SDI can be expressed in accordance with the BIS index.

Figure 6 The flow chart of ensemble artificial neural network (EANN) model construction.

Figure 7 One case demo of SDI.

(A) shows one SDI curve derived from a case ECG data, (B) one is the corresponding EACL, in which the blue thick line is the average of other five doctors’ thin lines.

Figure 8 Histogram distribution of correlation coefficient between SDI and EACL.

Except one in negative correlation, others are positive values, of which most are located at high value section from 0.6 to 0.9.

Table 2 The correlation coefficient comparison between EACL and both original SDI and ANN fitting SDI of 20 cases.

The latter one has better performance except few cases. From p value (paired Student t test), the two groups are considered statistically significant. (P < 0.05 means statistically significant).

Case	Original SDI & EACL	Fitting SDI & EACL	
1	0.7456	0.8478	
2	0.8263	0.8799	
3	0.8756	0.9570	
4	0.8812	0.9661	
5	0.7752	0.8857	
6	0.6732	0.7146	
7	0.7078	0.7197	
8	0.7818	0.7976	
9	0.7764	0.8880	
10	0.8400	0.9401	
11	0.8397	0.8815	
12	0.5817	0.6448	
13	0.7833	0.7330	
14	0.8585	0.9199	
15	0.9073	0.8764	
16	0.8445	0.8718	
17	0.6938	0.7565	
18	0.7736	0.8939	
19	0.8994	0.9198	
20	0.7902	0.922	
Mean ± std	0.7928 ± 0.0830	0.8508 ± 0.0913	
p-value	0.0420	

ECG analysis

Data from the 110 participants were analysed to obtain the SDI. For every case, the SDI was calculated using data from the entire operation procedure, including the awake, induction, maintenance, and recovery states. All data were obtained under different types of anaesthetics to guarantee compatibility, and parameters were selected empirically. Because Dn was in the range of 0 ms to 0.5 ms, it was used as the length of the histogram. The number of cells used was 100–500, and the best performance was obtained for 250. Dividing the data range into 250 cells required a cell width of 0.002, and the data block, M, was set as 128. Sample frequency Dn of 4 Hz was used, and thus one data block required 32 s. At any one time, 64 s of data (two 32 s data blocks) were required to calculate the SDI.

ANN analysis

The Pearson correlation coefficient was calculated for 110 intact cases. To measure the DoA accurately, regression analysis was conducted to compute the model. ANN analysis was utilised to determine the relationship between the SDI and EACL, thereby generating a more accurate output for evaluation. An ANN consists of three parts: an input layer, a hidden layer, and an output layer. In this study, a feedback propagation–type ANN was used, which is the most widely used type of ANN in machine learning. In previous studies (Huang et al., 2013; Liu et al., 2015; Sadrawi et al., 2015), nonlinear and nonstationary medical data were used with a back propagation network that had four layers: an input layer, two hidden layers with 17 and 10 neuron nodes, respectively, and an output layer. The number of nodes and layers used is widely known to affect the performance of an ANN, including the fitting effect and time elapse. From an engineering perspective, three to four layers are mostly used (Kourentzes, Barrow & Crone, 2014; Ripley & Ripley, 2001). In this study, different ANN topologies were tested, where the performance of the network varied as a function of the data type. A final topology was selected that obtained the highest accuracy in the shortest time.

Because the SDI data series is being used as the input to obtain a result similar to the EACL, the SDI needed to be consistent with the EACL for each case. As previously mentioned, there were variations in the subjective opinions of the five anaesthetists who completed the EACL, which thus resulted in a low correlation coefficient due to the different assessments. Therefore, 105 out of 110 data sets that had correlation coefficients higher than 0.3 (most of the value distribution was much higher than 0.3, as shown in the following ‘Statistical distribution of the correlation coefficient’ were used for ANN regression. In addition, 85 data sets were used separately in the model’s construction: 70% were used for training, 15% for validation, and 15% for testing. To enable selection of the best neural network, 1,000 epochs were set, and a large volume of data was employed to guarantee that the ANN model had a favourable fit. After the ANN model was generated, 20 sets of data were used for pure-testing of the ANN model to validate its performance.

The modelling procedure was repeated 10 times to generate 10 ANN models for cross-validation, and the procedure involved was as follows. The initial weights were set randomly, and as mentioned previously, the training was set to 1,000 epochs. The data were finally used to create 20 models for testing of model accuracy. The data were acquired from regular surgical procedures conducted in the NTUH using valid and strict operating procedures and identical regimes. Each model was totally different, due to the randomness of the initial weights. The performance for the cross-validation of 10 models was then calculated to check the variability of the ANN models. The results showed that a different model was created each time ANN training was performed, despite using the same data set for the training, validation, and testing. Cross-validation was conducted in a blind test to prove that there was no change in the regression result despite changes in the samples input.

In addition, an EANN was employed to optimise the prediction results. Utilisation of an ensemble obtains higher accuracy than using other neural network approaches (Minku & Yao, 2012) and can address the trade-off between prediction diversity and accuracy within an evolutionary multiobjective framework (Chandra & Yao, 2004). As shown in Fig. 6, a single network model can be established with the random creation of initial weights, scales, and parameters. In this study, 85 data cases were used to generate 10 ANN models with different initial weights, and the 10 ANN outputs were then averaged to validate the 20 cases for optimising the regression effect. Because each ANN generates a different result with a different error, the average of the model outputs was calculated to overcome associated errors, thus creating an EANN to improve results. All data analyses were conducted with MATLAB (Mathworks, R2014b, US).

Figure 9 Difference between the original SDI and fitting SDI for correlation coefficient, mean absolute error (MAE) and area under curve (AUC).

All of them (A) Correlation Coefficient; (B)Mean Absolute Error and (C) AUC indicates that the fitting SDI has better performance.

Table 3 The MAE between EACL and both original SDI and ANN fitting SDI of 20 cases.

The latter one shows better performance except in a few cases. From p value (Paired Student t test), the two groups are considered statistically different indicating the good ANN fitting results. (P < 0.05 means statistically significant).

Case	Original SDI & EACL	Fitting SDI & EACL	
1	25.3235	2.9221	
2	24.4898	3.1145	
3	24.4483	8.9847	
4	21.6974	4.6953	
5	38.0500	6.3051	
6	8.6140	9.0382	
7	46.8434	11.4393	
8	30.7200	4.5732	
9	23.8712	6.0356	
10	41.8986	14.1500	
11	36.0559	3.1404	
12	35.9865	3.5006	
13	33.9785	5.3338	
14	28.5371	5.0643	
15	33.0614	9.2370	
16	22.8827	4.0254	
17	33.6476	7.6811	
18	29.6125	9.1065	
19	19.8529	3.5845	
20	36.3620	4.3487	
Mean ± std	29.7967 ± 8.7180	6.314 ± 3.1201	
p-value	9.2214e−14	

Table 4 The AUC between EACL and both original SDI and ANN fitting SDI of 20 cases.

P value (Paired Student t test) show two groups are significantly different. The latter one has higher mean value and lower standard deviation. (p < 0.05 means statistically significant).

Case	Original SDI & EACL	Fitting SDI & EACL	
1	0.9493	0.9985	
2	0.8805	0.9771	
3	0.8992	0.9973	
4	0.9013	0.9999	
5	0.8272	0.9229	
6	0.6574	0.8843	
7	0.7386	0.8800	
8	0.5786	0.8181	
9	0.9691	0.9692	
10	0.9781	0.9878	
11	0.9926	0.9557	
12	0.9990	0.9213	
13	0.9575	0.9120	
14	0.8326	0.9892	
15	0.7216	0.9141	
16	0.9059	0.9520	
17	0.9876	0.9874	
18	0.8992	0.9993	
19	0.8508	0.9921	
20	0.9408	0.9924	
Mean ± std	0.8733 ± 0.1176	0.9525 ± 0.0510	
p-value	0.0088	

Statistical analysis

Statistical analysis was performed using SPSS (IBM v22, North Castle, NY, USA) and MATLAB. To evaluate the ANN effect, the performance of the original SDI was compared with the one random ANN regression–derived SDI. The Pearson correlation coefficient, mean absolute error (MAE), and area under the curve (AUC) for the EACL were computed and considered the gold standard. The receiver operating characteristic (ROC) curve was calculated to obtain the AUC, which is often used in medical fields during diagnosis of disease. The binary threshold used to distinguish between anaesthesia and consciousness was set to 65 (Johansen & Sebel, 2000). The parametric paired Student’s t-test was then used to assess the statistical significance. To prove the capability of the EANN-derived SDI to measure DoA, its relationship with EACL was analysed. Furthermore, the commonly used BIS was used as a reference. The same significance test was also undertaken between the two indices, thus demonstrating a solid and convincing result.

Results

Demonstration of typical SDI pattern

Figure 7A shows a typical SDI trend for a representative patient, and Fig. 7B displays the corresponding EACL obtained from the scores of five experienced and professional anaesthesiologists. The DoA is shown to change throughout the operation, where a higher value denotes a lower level of consciousness. After induction, the SDI falls sharply, although some variation exists in the maintenance period, and the SDI increases dramatically during emergence from anaesthesia. Generally, it corresponds with the fluctuations of EACL.

Statistical distribution of the correlation coefficient

To determine the coefficient distribution characteristics of all 110 data sets, a histogram with a cell width of 0.1 was constructed (Fig. 8). Most of the data values are located in the range from 0.6 to 0.8, with mean ± SD equal to 0.78 ± 0.16, which reflects a strong relationship with the EACL. Only five cases show extremely low correlation, these cases were just discarded.

Comparison between performance of original SDI and SDI fit using an ANN

An ANN model can be trained to model nonlinear behaviour and was used to accurately evaluate DoA in this study. Twenty data sets were used to quantify the optimisation effect, and a comparison was then made to validate the ANN effect. The correlation coefficients between the EACL and both the original SDI and ANN-derived SDI for cases 1 to 20 are presented in Table 2. It is evident that the ANN-derived SDI has significantly improved correlation with the EACL compared with the original SDI (p < 0.05). From the mean value of the statistics shown in Fig. 9A, it is clear that the ANN-derived SDI has superior performance. Table 3 compares the MAE results in the shape of correlation coefficients. The MAE fitting results obtained for the ANN-derived values are much smaller than those obtained without the ANN, which demonstrates that the ANN performed favourably. It decreases the difference much from the EACL by showing the statistical results in Fig. 9B significantly (p < 0.05). In addition, the AUCs of both the original SDI and the ANN-derived SDI for the 20 cases were calculated, and the results are shown in Table 4. Furthermore, the ROC curve for one case is presented in Fig. 10 and proves that the optimised SDI evaluates the level of consciousness more accurately. Figure 9C shows that the AUC of the ANN-derived SDI is 0.95 ± 0.05, much higher than that of the original SDI. The paired Student t-test was then used to determine the difference level between the two groups. The comparison reveals a statistically significant difference (p < 0.05), indicating the favourable fitting effect for the SDI using the ANN. From the relationship and the value difference, it is evident that the ANN-derived SDI measures the DoA more accurately than the original SDI.

Figure 10 The receiver operating characteristic (ROC) curve of original SDI and artificial neural network (ANN) derived one.

Both show the prediction of DoA features (AUC > 0.5). The ANN fitting SDI (blue curve) has larger AUC than the original SDI (red one), indicating better ability to predict DoA.

Figure 11 One typical representative of the ANN regression effect for SDI.

The blue line represents the ANN derived output; it has more similar fluctuation rhythm with EACL (black line). Relatively, the original SDI (red line) shows weaker relationship.

A typical ANN-derived curve is displayed in Fig. 11; the results were derived from the case shown in Fig. 7. Clearly, the ANN-fitted SDI is superior to the original SDI, which varies sharply at the induction stage, whereas the ANN-derived SDI is basically consistent with the EACL. Furthermore, the original SDI reaches zero during the early maintenance period, which is definitely unreasonable.

ANN blind cross-validation

The results detailed demonstrate that the ANN model improves the SDI performance. However, because only one ANN model test was conducted, a blind cross-validation test was conducted using the previously mentioned 20 cases to ensure that the ANN model was efficient. The results are presented in Table 5 and reveal that all 10 ANN models used for the 20 cases provide similar results. The same validation test was used for the MAE (Table 5). This demonstrated that the samples selected do not affect the construction and effectiveness of the ANN.

Table 5 The correlation coefficient and MAE (mean ± std) between 10 group ANNs fitting SDI and EACL of 20 cases.

From the mean value comparison, it proves the ANN performance regardless of different input case data.

Case	Correlation coefficient	MAE	
1	0.8508 ± 0.0913	6.314 ± 3.1201	
2	0.8346 ± 0.0952	4.8873 ± 1.9292	
3	0.8417 ± 0.1025	5.8552 ± 2.6317	
4	0.8378 ± 0.0972	5.1737 ± 2.2588	
5	0.8398 ± 0.0945	4.9005 ± 2.1774	
6	0.8459 ± 0.0933	4.9101 ± 2.1289	
7	0.8448 ± 0.0921	4.8997 ± 2.2364	
8	0.8158 ± 0.0976	6.0248 ± 2.5059	
9	0.8340 ± 0.0959	5.4458 ± 2.4640	
10	0.8507 ± 0.0899	5.5916 ± 2.5198	

EANN-derived SDI compared with the BIS

To further improve the regression performance of the ANN, an EANN was utilised to predict the DoA. Figure 12A shows that the ANNs had little variance in terms of the correlation coefficient. The EANN has the highest correlation and the lowest standard deviation, thereby proving the superior performance of the EANN. In addition, the MAE distribution is shown in Fig. 12B. The individual ANNs had similar characteristics. In addition, the EANN has the lowest MAE, which is consistent with the correlation coefficient results.

In comparison with the commonly used BIS, Fig. 13 shows that the EANN-derived SDI performs better than the BIS evaluation when referring to the EACL as the gold standard. Differences in terms of the correlation coefficient, MAE, and AUC are all significant (p < 0.05 parametric paired Student’s t-test). We also chose one representative case for which to plot the ROC curve for both the BIS and EANN-derived SDI (Fig. 14), where the AUC illustrates better discrimination between anaesthesia and consciousness. Tables 6 and 7 provide detailed results for the EANN and BIS over 20 cases, respectively.

Figure 12 The mean value and standard deviation statistics of ANNs and the EANN.

(A) correlation coefficient; (B) mean absolute error. (A) shows that the ANN has little fluctuation difference regardless of input training data in terms of correlation coefficient. The EANN has the highest correlation with lowest standard deviation to prove the better performance of EANN. MAE distribution is given in (B). As to individual ANN, they have similar ability, but not significantly. Similar to the result of correlation coefficient, EANN has almost the lowest MAE.

Figure 13 Difference between the BIS and EANN derived SDI for correlation coefficient, MAE and AUC using EACL as gold standard.

(A) means correlation coefficient, (B) denotes MAE and (C) shows AUC; all of them indicate the EANN derived SDI behaves better. Asterisk * represents the significant difference (p < 0.05, parametric paired student test).

Figure 14 The ROC curve of BIS and EANN derived SDI from the representative case using EACL as gold standard.

Both show good capability of DoA prediction (AUC > 0.5). The EANN derived SDI (blue curve) has larger AUC than the BIS (red one), indicating better performance.

Table 6 The correlation coefficient and MAE value between EACL and EANN fitting SDI of 20 cases.

Compared with all single ANN performance in Tables 4 and 5, the mean of correlation coefficient of 20 cases here is higher with lower standard deviation, while the MAE also proves this with lower mean and standard deviation, meaning that the EANN perform better than just one single ANN.

Case	Correlation coefficient	MAE	
1	0.8413	2.1975	
2	0.8871	3.1593	
3	0.9497	6.8287	
4	0.8994	4.6681	
5	0.8404	6.1740	
6	0.8081	4.3851	
7	0.7286	8.0616	
8	0.8704	3.4809	
9	0.8799	3.1161	
10	0.9411	2.3909	
11	0.8477	2.9354	
12	0.7722	4.9511	
13	0.7716	4.7145	
14	0.9041	3.4764	
15	0.8736	6.4892	
16	0.8848	8.3562	
17	0.7667	3.5179	
18	0.8385	6.5030	
19	0.9127	2.4303	
20	0.9145	3.1895	
Mean ± std	0.8566 ± 0.0612	4.5513 ± 1.9049	

Table 7 The correlation coefficient, MAE value and AUC between EACL and BIS of 20 cases.

These results are used to make comparison with EANN derived SDI. Significance test results are shown in Fig. 13. Generally, the BIS has weaker evaluation of DoA compared to EANN derived SDI in Table 6.

Case	Correlation coefficient	Mean absolutely error	AUC	
1	0.7746	7.5005	0.9951	
2	0.7798	4.9937	0.8878	
3	0.621	17.7697	0.7919	
4	0.3891	10.4033	0.9423	
5	0.8188	6.4099	0.9995	
6	0.555	20.6271	0.8773	
7	0.7116	14.7956	0.9031	
8	0.5617	6.1885	0.8036	
9	0.574	9.7251	0.9884	
10	0.7187	8.7184	0.9848	
11	0.6139	8.8011	0.9703	
12	0.694	9.9009	0.9302	
13	0.6949	12.3573	0.976	
14	0.6507	7.5062	0.996	
15	0.5636	10.4242	0.861	
16	0.663	8.0178	0.9758	
17	0.8089	7.4653	0.9815	
18	0.8937	8.8475	0.9942	
19	0.7989	5.8428	0.9914	
20	0.7553	9.0309	0.9782	
Mean ± Std	0.6821 ± 0.1164	9.7663 ± 3.8673	0.9414 ± 0.0637	

Discussion

Doctors use many observations and physiological vital signs to evaluate level of consciousness during clinical operations. The medical parameters are usually HR, BP, and photoplethysmography (Merry et al., 2010). However, because these parameters cannot accurately represent the actual DoA, researchers have been developing new methods for this purpose. For example, auditory evoked potential (AEP)- and EEG-related indices (which are mentioned in ‘Introduction’) such as BIS or entropy have been employed to quantify DoA (Liu et al., 2015; Nishiyama, 2013; Rosow & Manberg, 2001), and such indices are powerful and effective to some extent. An SDI method, which is based on ECG signals, is proposed in this study to measure DoA. The SDI method has a strong relationship with HRV, which is correlated with autonomic nervous system (ANS) function. Such function is seriously affected by anaesthesia (Hsu et al., 2012; Tarvainen et al., 2010), and because this fact is widely accepted in the field of anaesthesia, the ECG has often been used in DoA research.

Our aim was to construct a practical ECG-derived index, and thus the SDI proposed in this study is constructed to correspond with the EACL, the gold standard that researchers adhere to when developing methods of measuring DoA. EACLs were thus obtained by our research team members, which involved a large amount of effort and endeavour. Although DoA was clinically scored by experienced anaesthesiologists in this study, there were limitations associated with the subjective opinions of each anaesthesiologist, and it was thus necessary to collaboratively score certain cases. The aim of this paper was to propose the use of the SDI to measure DoA; thus, the SDI still requires certain future improvement with respect to the mathematical principles used. For example, the SDI is affected by ECG data fluctuations, which are related to the distribution and similarities between data block points. Parameter selection details must also be further investigated. Moreover, it is necessary to obtain a clearer understanding of the comparisons made between the SDI and the BIS, AEP, or entropy. It is considered that both EEG-derived and ECG-derived indices provide specific and useful features, and therefore further research is required in this respect.

The ANN regression model used herein was obtained from a predefined framework of an initial neural network based on our previous engineering research experience (Jiang et al., 2015; Liu et al., 2015; Sadrawi et al., 2015). However, it would be beneficial to investigate the ANN’s parameters, such as numbers of layers, number of nodes in each layers, and type of ANN (Hinton et al., 2012), and to discuss the weights and expiration criteria for the maximum optimisation of the performance.

Mathematically, the SDI does not represent heart rate or HRV but quantifies the difference between two consecutive data blocks (as explained in detail in ‘Materials and Methods’). When the difference is higher, the SDI value is also higher. The index is presumably affected by the shape of the distributions, as well as their similarity. If P1 and P2 are identical but both show either a uniform distribution (each value equally likely) or are deterministic (only a single value occurs in both), for example, different SDI will result. In the latter case, the SDI =1 − 12 = 0, and in the former case, SDI =1 − 100 × (1∕100)2 = 0.99, for n = 100. Therefore, the SDI not only measures similarity but is also affected by D(n), which means it can represent ECG data variability. Instead of simply using the correlation coefficient between the ECG and EACL as a definition of the SDI, which would be less dependent on shape, we used the procedure outlined in ‘Data analysis’, part A, to define the final standard SDI. Although an ANN has a relatively complicated relationship with DoA, it is utilised for the regression and an output is obtained to quantify DoA, thus solving the nonlinearity between the SDI and DoA. In addition, when patients are conscious, the ANS has a regulation function that affects ECG signals. Certain types of heart disease influence HRV (Mazzeo et al., 2011) and probably also the SDI. It is thus urgent for us to validate and optimise this potential effect, even though the regression results appear to be favourable. We do, however, assume that the SDI is not currently suitable for use in all occasions, and research is thus required to explore and amend any problems with the algorithm.

Although data from more than 100 cases were collected to build the SDI and the results demonstrate favourable performance, most of our cases were middle-aged patients. Therefore, it is necessary to obtain more data from young patients to verify our methodology (Cornelissen et al., 2015; Gemma et al., 2016), Surgery is conducted with respect to certain protocols and patient safety is always the priority; therefore, the anaesthetic drugs used for the patients in this study were all chosen by experienced anaesthesiologists, who perhaps favoured the use of particular drugs. Although other types of drugs could also deliver successful outcomes (Mawhinney et al., 2012; Schwartz et al., 2010), the data obtained during the maintenance period were only related to the administration of propofol, sevoflurane, and desflurane (Table 1). It is thus necessary for us to obtain data based on the use of other drugs such as medetomidine, isoflurane, and nitrous oxide (Kenny et al., 2015; Purdon et al., 2015), which may enhance index compatibility. In addition, mixed anaesthetic agents were given to the patients, which made it difficult to evaluate the capability of the SDI to reflect the use of one specific drug regime. Furthermore, our data are obtained from routine surgery performed in a hospital and do not involve any other clinically specific anaesthetic settings; thus, investigations of this aspect would also be useful. We will conduct future experiments using related data, and strict and rigorous comparisons will be made between indices. Future efforts will be made to investigate and update our algorithm and to determine the possibility of improving DoA evaluation accuracy through a combination with BIS or entropy, for example, or consideration of different surgical circumstances.

Another issue to be considered is the spectral analysis of the ANS. ANS function has been widely employed in the assessment of DoA using ECG frequency domain features (Guzzetti et al., 2015; Lin et al., 2014). Previous articles have mainly focused on the ratio between high and low frequency powers. Galletly et al. (1994) described the spectral influence of several common anaesthetic agents on HRV, which provides directions for spectral part analysis. In addition, multitaper time frequency analysis was undertaken for autonomic activity dynamics evaluation in Lin et al. (2014). Nevertheless, future research on spectral analysis is required to pursue the promising and valuable integration with the present temporal analysis. Finally, although the results of this work symbolise DoA from the perspective of the ANS, we also aimed to provide an alternative to EEG-derived evaluation (Purdon et al., 2015; Samarkandi, 2006; Sleigh & Barnard, 2004). Based on the results of this research, it is considered that to overcome the disadvantages of EEG-based methods, studies should be initiated using a combination of EEG- and ECG-based methods.

Conclusions

In this study, physiological data from 100 participants were analysed to determine the ability of our SDI algorithm to evaluate DoA. ECG data were used to derive the SDI, representing the differences in HRV to demonstrate the ability of the SDI to measure DoA. To optimise prediction accuracy, ANN models were constructed and blind cross-validations were performed to conduct a regression test. In addition, an EANN was employed to overcome random errors and overfitting of the ANN models. This study indicated that HRV analysis has the potential to become another effective method for the evaluation of DoA. However, because there is a current lack of ideal measurement methods for the assessment of patient consciousness level, it is considered that incorporating the SDI into other methods would be useful. Therefore, combining the use of the SDI with other physiological medical signals relating to anaesthesia, such as EEG signal, would also be meaningful and helpful in improving the accuracy of DoA evaluation.

Additional Information and Declarations

Competing Interests

Author Contributions

Human Ethics

Data Availability

The authors declare there are no competing interests.

Quan Liu analyzed the data, prepared figures and/or tables.

Li Ma performed the experiments, analyzed the data, contributed reagents/materials/analysis tools, wrote the paper, prepared figures and/or tables.

Ren-Chun Chiu performed the experiments, analyzed the data, contributed reagents/materials/analysis tools, prepared figures and/or tables.

Shou-Zen Fan conceived and designed the experiments, performed the experiments, analyzed the data.

Maysam F. Abbod performed the experiments, wrote the paper, reviewed drafts of the paper.

Jiann-Shing Shieh conceived and designed the experiments, wrote the paper, reviewed drafts of the paper.

The following information was supplied relating to ethical approvals (i.e., approving body and any reference numbers):

All studies were approved by the Research Ethics Committee, National Taiwan University Hospital (NTUH), Taiwan, and written informed consent was obtained from patients (No: 201302078RINC).

The following information was supplied regarding data availability:

Ma, Li; Liu, Quan; Chiu, Ren-Chun; Fan, Shou-Zen; Abbod, Maysam F.; Shieh, Jiann-Shing (2017): Raw Data.rar. figshare.

https://doi.org/10.6084/m9.figshare.5254426.v1.

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
