# Peer review of "HRV-derived data similarity and distribution index based on ensemble neural network for measuring depth of anaesthesia"

_PeerJ, doi:10.7717/peerj.4067_

## Round 0.1 · original submission · Major Revisions

The authors need to clarify the method section, compare to EEG-based measures (BIS or entropy) and check the English writing.

·

Basic reporting

This article introduces a heart rate variability based method to measure anesthetic depth.

In the introduction section, authors stated that EEG based indices are not suitable for deph of anesthesia (DoA) evaluation. They also stated that HRV would be more suitable to monitor DoA. To my own knowledge EEG based indices like BIS or GE entropy are considered as gold standards and are widelly used worldwide for DoA evaluation. I think authors should provide a more exhaustive literature references list to affirm that EEG is not suitable for DoA monitoring.

In the discussion, authors cited Jeanne et al 2009. This paper is related to nociception ; not to DoA.

Experimental design

The research question is well defined, relevant and meaningful.

Though the research question is well defined, the material and method section need to be described with more details.

Regarding the anesthetic procedure ;

- How anesthesia was performed ? Did all the patients received gas (sevoflurane, desflurane,...) ? How hypnosis was maintained (propofol ? gas ?) ? How anesthetic agents were adapted ?

- Were EEG indices like BIS or entropy measured during anesthesia?

- Can authors list all the physiological/anesthesia parameters monitored during the procedure ?

Validity of the findings

The statistical analysis is not clear and should be completly rewrite ;

- How do authors justify the sample size ?
- Statistical analysis used for validation (ROC curve) are not clearly explain.

Additional comments

The statistical analysis needs to be explain more clearly.
I do not understand why authors didn't compared their method to the commonly use EEG based indices.

Reviewer 2 ·

Basic reporting

The English language needs to be substantially improved in order for the paper to be accessible to a broad, international audience. This is perhaps the biggest issue with the paper.

The literature review and structure of the paper is appropriate.

There are several details regarding the methods that are, however, opaque. This includes (i) the criteria by which the experts scored the level of consciousness (i.e., the EACL); (ii) the construction of the ensemble neural network; (iii) the criteria for artifact removal in the ECG time series.

Experimental design

The experiment design is sound, and it is clear that substantial effort was put into execution. In particular, the effort to deploy five different anesthesiologists to obtain expert labels is commendable. Also, the cross validation for the machine learning steps looks to be well designed. Overall, the study seems rigorous.

However, there are methodological details that should be provided in the text in order to make the paper results more reproducible. This includes being more systematic about the criteria used by the anesthesiologists to score the data; as well as providing better clarity on the design of the ensemble neural network (especially for readers who do not have a background in neural networks for classification applications).

Validity of the findings

The major contributions of the study are in the use of artificial neural networks to enable classification of depth of anesthesia from the so-called SDI, a measure of non-stationarity in the ECG time-series. Here, non-stationarity is assessed by obtaining an empirical distribution over sliding windows of the raw data, minus manually removed artifacts. I would encourage the authors to concisely state these contributions in the introduction of the paper, so they are clear up front to the paper.

As mentioned, I believe the results are overall interesting and sound. My major concern is whether the correlation coefficient (between the SDI/ANN-derived classifier and the EACL) is the best way to measure performance. From an application standpoint, what seems most important is determining when a patient is or isn't conscious, and thus I wonder why the authors didn't also do a more direct analysis on predicting these states in a more binary fashion (i.e., rather than attempting to obtain correlate against a continuous index). As well, in understanding the power of the method it would be very important to compare performance to other DoA measures, though I accept the authors' argument that this is best reserved for future work.

There are also key questions about how the performance varies with different anesthetic regimes. The authors provide the different types of drugs used, however do not disassociate the classifier performance in these terms. For example, to the poorly performing cases all correspond to one particular drug regime?

Additional comments

This is a well-designed study that highlights the potential utility of ANNs to classify ECG-derived biomarkers for DoA classification. However, since the main contribution of the paper is the use of the ANN, I think a more complete study may use other covariates in order to highlight why the SDI is the best one for this application; as well as how performance varies with different drug regimes.

The paper must be carefully copyedited in order for it to be accessible to a broad, international scientific audience.

---

## Round 0.2 · Minor Revisions

The reviewers raised a few remaining minor comments that need to be addressed before the paper can be accepted.

·

Basic reporting

English editing is OK.
Literature references is more exhaustive.

Experimental design

Method is more clear. Authors added details about the anesthesia procedure and the statistical analysis.

Validity of the findings

Authors added comparison with the EEG based indexes as requested.

Additional comments

The reference to Jeanne et al. has not been deleted. This reference is about HRV and analgesia ; not HRV and DoA.

Reviewer 2 ·

Basic reporting

The writing has been improved since the original submission.

There are some minor typos in the Figure captions that should be addressed (e.g., Figure 13 caption is truncated).

Experimental design

The experiment design is rigorous.

Validity of the findings

The findings appear valid and well-constrained. I appreciate the authors' contextualization of their results. The new figure 13 is helpful in understanding the merits of the proposed approach in contrast to BIS. While the gains are modest according to certain metrics, it is clear that the proposed approach is accomplishing the intended goal of predicting DoA.

Additional comments

I recommend revisiting the sentence on p4, line 82 motivating neural networks. We do not fully understand the mechanisms of ANNs and it is an overtstatement to say they operate similarly to manner to biological networks.

---

## Round 0.3 · accepted · Accept

The authors have appropriately addressed the remaining comments.